# The Effect of Cation Disorder on Ferroelectric Properties of Sr_x_Ba_1−x_Nb_2_O_6_ Tungsten Bronzes

**DOI:** 10.3390/ma12071156

**Published:** 2019-04-10

**Authors:** Solveig S. Aamlid, Sverre M. Selbach, Tor Grande

**Affiliations:** Department of Materials Science and Engineering, NTNU Norwegian University of Science and Technology, NO-7491 Trondheim, Norway; solveig.s.aamlid@ntnu.no (S.S.A.); selbach@ntnu.no (S.M.S.)

**Keywords:** SBN, TTB, strontium barium niobate, tungsten bronze, ferroelectricity, thermal history, cation disorder

## Abstract

The versatile crystal structure of tetragonal tungsten bronzes (A1_2_A2_4_C_4_B_10_O_30_) can accommodate complex stoichiometries including cations in different valence states and vacant cation sites. Here, we report on the effect of thermally induced cation-vacancy disorder in the tetragonal tungsten bronze Sr_x_Ba_1−x_Nb_2_O_6_ (SBNX). SBNX (x = 0.25, 0.33, 0.50, 0.61) ceramics, prepared by conventional solid-state synthesis, were annealed at varying temperatures and subsequently quenched to room temperature. The Curie temperature of all the SBNX materials increased with higher quenching temperatures, accompanied with ferroelectric hardening. The variation in thermal history also caused structural changes, specifically a contraction of the *a* lattice parameter and a minor elongation of the *c* parameter. These effects are discussed in relation to recent first principles calculations of the energy landscape of the cation-vacancy configurations and experimental evidence of thermally induced cation-vacancy disordering.

## 1. Introduction

The tetragonal tungsten bronze (TTB) structure has the general formula A1_2_A2_4_C_4_B_10_O_30_, where the two A1 and A2 cation sublattices have sites of relatively similar size. The largest A1 and A2 sites accommodate alkali, alkaline earth, lanthanides, and lone-pair cations, the smaller B site can host high-valence transition metal cations, and the smallest C site is usually empty, but can be occupied by lithium or other small cations [1]. The crystal structure is versatile with respect to composition, and one of the most studied ferroelectric TTBs is the solid solution Sr_x_Ba_1−x_Nb_2_O_6_ (SBNX, *X*
∈(0.25<X<0.75) [2]. The Curie temperature of the paraelectric to ferroelectric phase transition is known to decrease as the transition gradually becomes more relaxor-like with increasing strontium content, from a true ferroelectric for low strontium content and typical relaxor behavior for high strontium content [3]. The mechanism for the relaxor behavior in SBN is debated; it has recently been shown that incommensurate modulations accompany relaxor behavior in many tungsten bronzes, but in the case of SBN, the incommensurate modulation is present regardless of the barium content and relaxor properties [1]. The lowering of the Curie temperature is accompanied by a decrease of the $a/c\sqrt{10}$ ratio, which is analogous to tetragonality in perovskite structures. The sintering behavior of SBN is reported to be composition dependent due to the convex shape of the solidus line in the phase diagram [4], and it is challenging to control the microstructure due to abnormal grain growth [5].

SBN is an unfilled tungsten bronze, meaning that the A1 and A2 sublattices are occupied by a mixture of strontium, barium, and cation vacancies [6]. The similar size of the A1 and A2 sites opens the possibility of different local ordering of cations and vacancies, and entropy-driven vacancy disordering. A recent computational study of the end components of SBN, strontium niobate (SN) and barium niobate (BN), revealed that the different cation/vacancy configurations are close in energy, making especially the Ba-rich compositions sensitive to temperature changes [7]. Considering the similar size of the A1 and A2 lattice sites, the fact that Ba is known to occupy A1 sites in other TTB compounds such as Ba_4_K_2_Nb_10_O_30_ (BKN) [8] and the flat energy landscape of BN from DFT [7], we would expect Ba to be found also on the A1 site in SBN a priori. However, several diffraction studies concluded that Ba will only occupy the larger A2 sites in SBN [6,9,10].

The effect of cation ordering is well known in spinels, and experimental data on cation disorder have successfully been described by a thermodynamic model proposed by Navrotsky and O’Neill [11]. Similar order-disorder is anticipated also to occur in TTB. Already in 1969, the effect of quenching Ba_4_Na_2_Nb_10_O_30_ (BNN) from high temperatures was studied, resulting in a change in Curie temperature of 33 °C attributed to cation disordering [8]. A similar study was later performed on SBN, where the effect of a higher quenching temperature was the opposite of what was seen in BNN, specifically that the Curie temperature increases with more disorder [12]. Recently, the structural changes in SBN53 after quenching from higher temperatures were reported, showing a redistribution of the vacancies from the A2 to the A1 site, changes in the lattice parameters and structural modulations [10]. Finally, oxygen vacancies, which form during annealing at elevated temperatures, may enhance the thermoelectric properties of SBN [13,14].

Here, we report on the effect of thermal history on four SBN compositions with special focus on SBN33. The thermal history of the sintered SBN materials was varied systematically by annealing and subsequent quenching to freeze in a specific cation distribution. Dielectric spectroscopy was applied to determine the Curie temperature as a function of thermal history. Ferroelectric hysteresis loops were also characterized for SBN33 ceramics with different thermal histories. Rietveld refinements of X-ray diffraction (XRD) patterns were performed to study changes in the lattice parameters and Ba/Sr population of the A1 and A2 sublattices. The present findings are discussed with respect to possible cation order/disorder in SBN and dielectric loss due to the thermal reduction of Nb.

## 2. Materials and Methods 

Sr_0.25_Ba_0.75_Nb_2_O_6_ (SBN25), Sr_0.33_Ba_0.67_Nb_2_O_6_ (SBN33), Sr_0.50_Ba_0.50_Nb_2_O_6_ (SBN50), and Sr_0.61_Ba_0.39_Nb_2_O_6_ (SBN61) materials were prepared by solid state synthesis with SrCO_3_ (>99.9%), BaCO_3_ (>99.98%), and Nb_2_O_5_ (>99.99%) precursors (Sigma Aldrich, Darmstadt, Germany). Stoichiometric amounts of the dried solid precursors were mixed in ethanol, dried, uniaxially pressed into pellets (30 MPa), and calcined for 8 h at 1100 °C. The calcined pellets were crushed to powder in a mortar, mixed with ethanol, and ball milled for 20 h using yttria stabilized zirconia milling balls. The dried powder was sieved through a 250-µm sieve and uniaxially pressed to pellets 1 mm thick and 10 mm in diameter (90 MPa). The pellets were further compacted with a cold isostatic press (200 MPa).

Solid state sintering of the samples was performed in air at 1400 °C for 12 h for SBN50 and SBN61, 1375 °C for 12 h for SBN25, and 1350 °C for 4 h for SBN33, all with a heating and cooling rate of 300 °C h^−1^. The sintering conditions for the four compositions were optimized to obtain samples with sufficiently high density for characterization of electrical and dielectric properties. The surface and edges of the pellets were polished away to ensure that abnormal grain growth or coarse microstructure on the surface of the pellet did not influence the electrical measurements. Annealing of the sintered materials was done in air with a heating rate of 300 °C h^−1^ to either 800, 1000, 1200, or 1350 °C with a soaking time of 30 min. The sample was then removed from the furnace and allowed to cool in air. The SBN25 and SBN33 samples heat treated at 1200 °C and 1350 °C were also heat treated at 500 °C for 4 h in air and are referred to as reoxidized in the following. The microstructure of the sintered samples was characterized by scanning electron microscopy (S-3400N, Hitachi, Japan), after polishing with diamond suspensions with decreasing particle size down to 1 µm. The samples were thermally etched for 5 min at 1225 °C in air and sputter coated with Au before imaging.

Dielectric spectroscopy was performed using an Alpha-A impedance analyzer (Novocontrol, Montabaur, Germany) in a Novotherm (Novocontrol, Montabaur, Germany) testing chamber, allowing precise temperature control (±0.5 °C). The dielectric properties were continuously measured every decade from 1–10^6^ Hz with a heating rate of 2 K min^−1^. The temperature error was about 1 °C, as the measurement of the full frequency range took about 30 s. The analysis was performed on data from the cooling cycle. The samples were prepared by grinding to a flat parallel geometry using 1200 SiC paper and covered in a silver paste for the electrode, or sputtered gold in the case of SBN33. 

Ferroelectric hysteresis measurements were performed on the SBN33 ceramics with gold sputtered electrodes, using the Piezoelectric Evaluation System aixPES (aixACCT, Aachen, Germany). The samples were measured at 1, 10, 50, and 100 Hz, with steps of 200 V/mm. 

Synchrotron XRD on the samples heat treated at 800 °C was carried out at the Swiss-Norwegian beamline (SNBL, BM01) at the European Synchrotron Radiation Facility (ESRF). A LaB_6_ powder sample was used for calibration, and a structural model with space group *P4bm* for SBN61 reported by Carrio et al. was used as a starting point for Rietveld refinement with the Bruker AXS Topas 5 software [6]. The lattice parameters, atomic positions, site occupancy, and thermal parameters of the cations were refined. The overall composition was constrained to the nominal composition. Sr and cation vacancies could move freely between A1 and A2 sites, while Ba was fixed to the A2 site. Attempts to allow Ba on the A1 site resulted in unphysical occupancies, and Ba was only allowed on A2 sites, supported by previous reports [6,9,10].

Powder XRD was carried out on the thermally-annealed samples using a Siemens D5000 instrument (Siemens, Karlsruhe, Germany) with monochromatic Cu Kα_1_ radiation, a step size of 0.008° 2θ, and variable counting time (VCT) over the course of 24 h. The structural data obtained from the refinements using synchrotron data for the four compositions were used as a starting point for these refinements. The same parameters were refined except the thermal parameters and the oxygen positions, which were fixed to the values determined from the synchrotron data.

## 3. Results

### 3.1. Density and Microstructure

The sintered samples were all dense (>95%) and phase pure within the detection limit of XRD; see Appendix A. The microstructures of the four different compositions are shown in Appendix A. It is worth noting that SBN50 and SBN61 possessed a considerably larger grain size relative to the two other compositions due to the higher sintering temperature. 

### 3.2. Ferroelectric Properties

The temperature dependence of the dielectric permittivity and loss of SBN33 after thermal annealing and subsequent quenching are shown in Figure 1. The data are shown for 1000 Hz, and similar results were observed for other frequencies. A maximum in the permittivity, which corresponds to the Curie temperature, was observed for all the SBN33 samples with different thermal histories. A shallow minimum in the dielectric loss was observed close to the phase transition, and the dielectric loss rapidly increased above the transition. The most pronounced effect of the thermal treatment was the shift in the Curie temperature to a higher temperature with increasing quenching temperature. T_C_ was shifted from 174 °C to 226 °C by increasing the quenching temperature from 800 °C to 1200 °C, corresponding to an increase of T_C_ of 48 °C.

Analogous changes in the Curie temperature as influenced by thermal history were obtained for the three other compositions as shown in Appendix A. A comparison of the dielectric permittivity at 1000 Hz for all the compositions heat treated at 800 and 1350 °C is shown in Figure 2a. The Curie temperature decreased as the Sr content increased, in accordance with earlier work [3]. The maximum in the permittivity was shifted to higher temperatures after annealing at 1350 °C for all four compositions. The change in Curie temperature as a function of heat treatment temperature for each composition is shown in Figure 2b, where the most pronounced effect was observed for SBN33 quenched from 1350 °C. The shift in T_C_ with increasing quenching temperature was similar to what has previously been reported for SBN53 [12]. The heat treatments did not affect significantly the frequency dependence of the permittivity and the relaxor properties. This is shown for SBN61, the most relaxor-like composition, in Appendix A. The dispersion in T_C_ with different frequencies was small and not affected significantly by the heat treatments. Similar behavior was observed for the three other compositions. A strong frequency dependence in the absolute value of the permittivity was observed for samples quenched from 1350 °C and to some degree also from 1200 °C, which is attributed to the high and frequency-dependent conductivity of these samples, as discussed further below. 

The ferroelectric hysteresis loops and currents at four different frequencies for SBN33 heat treated at 800 °C and 1350 °C are shown in Figure 3. The sample annealed at 800 °C had a typical ferroelectric behavior with only a weak frequency dependence of the hysteresis loop, with a larger coercive field with increasing frequency. This is evident from Figure 3a, which also demonstrates switching events that happened over wider fields for higher frequencies. The sample heat treated at 1350 °C showed an increase in the coercive field, as displayed in Figure 3b corresponding to ferroelectric hardening. The coercive field was less dependent on frequency, but the switching events occurred over a much wider field, as is evident in Figure 3b. The hysteresis loops at low frequency showed clear evidence of a leakage current.

### 3.3. Structural Properties

The effect of increasing annealing temperature on the crystal structure was determined by XRD. The *a* and *c* lattice parameters as a function of quenching temperature are shown in Figure 4. The *c* parameter was linearly dependent on the composition in accordance Vegard’s law, while a non-linear variation of the *a* parameter with composition was evident, in agreement with previous reports [9,15]. The *c* parameters increased subtly with increasing quenching temperature, while the *a* parameter contracted with increasing heat treatment temperature, in excellent agreement with Graetsch [10]. This led to a smaller $a/c\sqrt{10}$ ratio, which explains the increase in Curie temperature measured by dielectric spectroscopy. The non-linear dependence of the *a* parameter was also dependent on the quenching temperature, possibly reflecting variation in the cation occupancy on the A1 and A2 sites, as discussed further below.

The variations in the composition on the A1 and A2 sites were investigated by Rietveld refinement of the diffraction data. The refined data are summarized in Appendix A, and Appendix A shows a typical fit. The cation and vacancy population on the two sites for the samples heat treated at 800 °C for all the compositions are platted in Figure 5. Values extracted from the Rietveld refinements of the X-ray diffraction patterns of the annealed samples are presented in Appendix A. The refined cation/vacancy occupancies of the temperature series are shown in Appendix A. We do not observe any clear trends in this dataset.

### 3.4. Reoxidation

Dielectric permittivity measurements were also performed before and after reoxidation of the SBN33 ceramic heat treated at 1350 °C. The permittivity and ac conductivity at 1000 Hz are shown as a function of temperature before and after reoxidation in Figure 6a. Without reoxidation, the high conductivity overshadowed the dielectric properties, and accurate determination of T_C_ became challenging. The frequency dependence of the dielectric loss and AC conductivity at ambient temperature are displayed in Figure 6b. Note that the reduction in the conductivity by reoxidation resulted in a ten-fold decrease in the magnitude of the apparent permittivity by reoxidation. This apparent rise in permittivity was due to the conductivity induced by the thermal reduction of Nb (Reaction 2). The conductivity at low frequencies is decreased by several orders of magnitude by reoxidation, while the conductivity at high frequencies is not affected.

## 4. Discussion

Our main finding was the confirmation that thermal history has a strong effect on the electrical and structural properties of SBN. The thermal history can be seen as a processing parameter to tailor the ferroelectric and structural properties. This work confirms previous reports that T_C_ can be increased by quenching from high temperature [12]. We also observed that this is accompanied by hardening of the ferroelectric properties, which has never been reported before. The main influence on the crystal structure was a significant contraction of the *a* lattice parameter and an expansion of the *c* lattice parameter. This is also in accordance with what was recently reported by Graetsch [10]. The microscopic origin of these effects of thermal history can be discussed with relation to the reaction:
[Av](A_4_)Nb_10_O_30_ = [A_1+x_v_1−x_](A_4−x_v_x_)Nb_10_O_30_,(1)
where square and round brackets denote the A1 and A2 sites, respectively, A is Sr/Ba, and v is a vacancy. This corresponds to the movement of a vacancy from A2 to A1, as illustrated in Figure 7. Reaction (1) was supported by first principles calculations [7] and diffraction studies of SBN52 [10]. In the following, the present findings are discussed in relation to the distribution of cations on A1 and A2 sites and vacancy order/disorder. In addition, we will also discuss the influence of thermal reduction of Nb on the conductivity in particular, which to the best of our knowledge has not been addressed before with respect to thermal history for TTBs.

With increasing Sr content, the lattice parameters decreased, the vacancy moved from A1 to A2, and the Curie temperature decreased, as reported previously [3]. The compositional variation in the *c* parameter, shown in Figure 4, followed Vegard’s law and was not strongly dependent on cation distribution on the A1 and A2 sites. It was mostly influenced by the average cation size determined by the Sr content. On the other hand, the *a* lattice parameter had a strongly non-linear behavior because it was more influenced by the population of Sr and vacancies on the A1 and A2 sites. The cation population on the two sites depended on the Sr content, as shown in Figure 5a. It is interesting to note that the vacancy concentration on the A2 sites converged towards a constant value on increasing Sr content; the vacancy never completely moved to the A2 site as predicted by DFT [7]. This is because the temperature where complete site order was expected was too low for the cations to be mobile [12]. The Curie temperature decreased with smaller average A cation size because the octahedra tilt to optimize the coordination of the smaller cation, and this tilt coupled to and suppressed the second-order Jahn–Teller distortion, causing ferroelectric polarization [16]. Finally, the incommensurate modulations found in the *ab* plane in the SBN system may also contribute to the non-linear composition dependence of the *a* parameter.

When the samples were quenched from increasing temperatures, we observed a contraction of the *a* lattice parameters and an increase in T_C_, with SBN33 being the composition most influenced by thermal history. SBN33 has the lowest inherent configurational entropy due to the 1:2 ratio between A1 and A2 sites, and the most stable configuration is Sr and vacancy on the A1 site and only Ba on the A2 site. This means that if thermal treatment results in random distribution of vacancies, this will lead to the largest effect of disorder. The contraction of the *a* parameter was in accordance with the work by Graetsch [10] and is explained by the changes in the cation/vacancy populations. The increase in the Curie temperature was in accordance with the work by Guo et al. [12] and can also be explained by the changes in the cation/vacancy populations. It is interesting to compare the increase in T_C_ by increasing Ba content and the increase in T_C_ due to thermal history since both are connected to an increase in the population of vacancies on the A2 site and how this couples to the octahedral tilting. The more Ba-rich compositions should also have a stronger driving force for disordering from DFT calculations [7]. Ba is also less mobile than Sr [7], and it is expected that it is easier to successfully freeze in more disorder in Ba-rich compositions. The combined effect of the ground state configurational entropy and the Ba content is that the effect of the thermal history was strongest for SBN33 followed by SBN25, SBN50, and finally SBN61.

Oxygen vacancy formation, accompanied by a reduction of a transition metal, is a common cause of changes in properties in transition metal oxides when quenching from high temperature [18,19]. SBN33 heat treated at 1350 °C was measured before and after reoxidation, as shown in Figure 6. SBN can be reoxidized at temperatures around 500–600 °C, as shown by thermogravimetric analysis [13], but the cations are immobile up to ~800 °C. Oxygen vacancies and changes in the valence state of Nb in the case of SBN will influence the electric properties, causing the materials to be more conducting and to turn grey due to the increased concentration of electronic defects. Oxygen vacancies and reduction of Nb were also expected to result in chemical expansion of the crystal lattice in tungsten bronzes, as they do in perovskites [20], but our observations show a contraction of the total volume of the unit cell with increasing quenching temperature. The cation disordering is therefore suggested to be the dominating effect on the structural effect of the thermal history. The oxygen vacancies are the dominating defect for the electronic properties and are expected to affect the octahedral distortions at significant concentrations, but further investigations are necessary to elucidate the effect of oxygen vacancies in tetragonal TTBs.

A possible Kroger–Vink point defect reaction for thermal reduction of Nb is:
(2)$$O_{O}^{x}+2Nb_{Nb}=V_{O}^{••}+2Nb_{Nb}^{\prime }+\frac{1}{2}O_{2},$$
which describes how oxygen can exit the structure into the gas phase and leave an oxygen vacancy ($V_{O}^{••}$), which is charge compensated by reduction of Nb, resulting in either a localized electron in a 4d-orbital on the Nb site ($Nb_{Nb}^{\prime }$) or a delocalized electron in the conduction band [13]. Here, it is suggested that the electron is localized on Nb, but it is beyond the scope of this work to determine whether oxygen non-stoichiometry in SBN results in itinerant or localized electrons. The high conductivity of the partly-reduced samples dominates the dielectric measurements, and the Curie temperature could not be determined for some of the samples quenched from 1350 °C. When these samples were reoxidized, the conductivity was reduced by several orders of magnitude. The dramatic decrease in the AC conductivity at low frequencies by reoxidation (Figure 6b) resembled similar observations reported for 0.7 BiFeO_3_-0.3 Bi_0.5_K_0.5_TiO_3_ [21], where the oxidation state of Fe could be controlled by annealing in different O_2_ partial pressures. The significant conductivity at low frequencies after heat treatment without reoxidation can be rationalized by oxygen vacancies formed by the defect equilibrium in Equation (2).

Assuming that the mobility of charge carriers is not changed, this corresponds to an equally drastic reduction in the concentration of point defects responsible for enhanced conductivity prior to reoxidation. It is concluded that the thermal reduction described by Reaction (2) caused the high conductivity. The point defect equilibrium is reversible and can be reversed by annealing at a lower temperature without affecting the cation configuration. We could therefore characterize the effect of the thermal history despite the challenges with the onset of thermal reduction of Nb. Previous works have also shown that the microstructure affects the Curie temperature [5]. Enhanced diffusion along grain boundaries and cracks is possible (see the Appendix A), which may explain why the microstructure is important for the dielectric properties.

The absolute values of the permittivity were observed to change significantly with composition and heat treatment. The reason for the most dramatic increases in permittivity is attributed to thermal reduction of Nb and oxygen vacancies and increased conductivity for samples quenched from 1350 °C. However, for lower quenching temperatures, the absolute value of the permittivity decreased slightly with increasing heat treatment temperature. The permittivity was lower for the SBN50 and SBN61 samples relative to the two others, most likely due to the larger grain size for these two compositions. 

The hardening of the ferroelectric properties of SBN due to quenching from higher temperature was unexpected based on previous work on hardening in classical ferroelectric materials such as PZT [22,23] and lead-free BiFeO_3_ [24]. In both of these materials, quenching from a high temperature results in a softening of the ferroelectric properties. In contrast to this behavior, a hardening of the ferroelectric properties was observed for SBN due to the quenching from higher temperatures. The microscopic origin of this apparent hardening is not known, and further investigations are necessary to identify the origin; however, we should still point out that this is most likely related to a point defect as in PZT and BiFeO_3_. In the case of SBN, the most likely point defects could be related to either the cation-vacancy disorder or oxygen vacancies accompanying the thermal reduction of Nb. Finally, it is worth noticing that the difference in the Curie temperature will influence the observed coercive field since the harder sample was measured at a temperature further away from the transition temperature than the softer one.

## 5. Conclusions

We have performed a systematic study of the effect of thermal history on the structural and electrical properties of four compositions in the SBN system. Thermal history has a profound effect on both the Curie temperature, the ferroelectric properties, and the lattice parameters in the SBN system. The dependence on the thermal history was discussed with relation to vacancy order-disorder phenomena caused by the movement of cation vacancies from the A1 to A2 site, in agreement with recent experimental and computational studies. We conclude that the cation-vacancy order-disorder phenomena in SBN tungsten bronzes affects both the structural and functional properties of the material. Finally, we suggest that other tungsten bronze compositions should be susceptible to the same kind of changes and that heat treatment can be viewed as a processing parameter to tailor the functional properties of new lead-free ferroelectrics.

## Figures and Tables

**Figure 1 materials-12-01156-f001:**
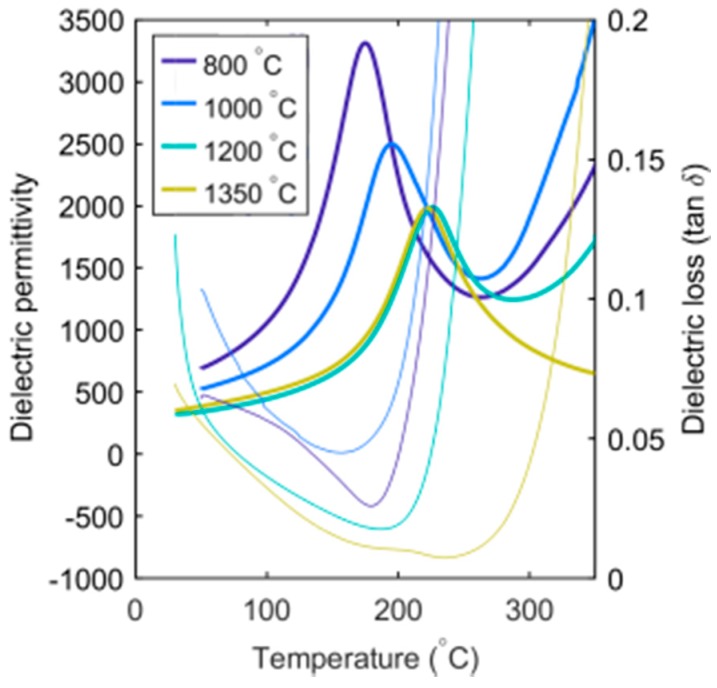
Dielectric permittivity (thick lines) and loss (thin lines) of SBN33 quenched from four different heat treatment temperatures at 1000 Hz. Note that the samples heat treated at 1200 and 1350 °C have been reoxidized.

**Figure 2 materials-12-01156-f002:**
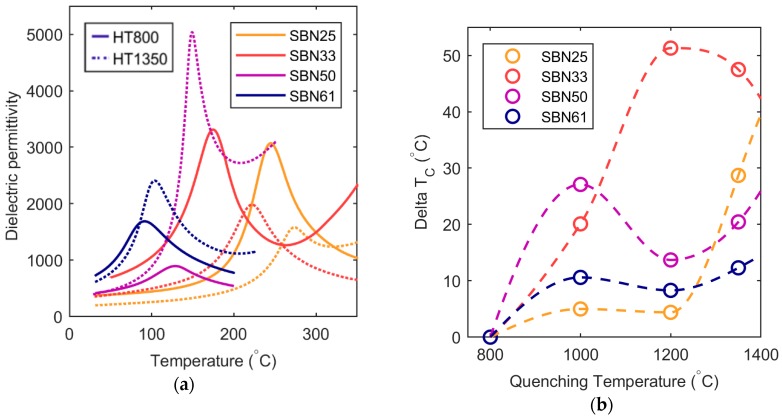
(**a**) Selected permittivity curves for all the compositions and two heat treatment (HT) temperatures. Note that SBN33 heat treated at 1350 °C have been reoxidized. (**b**) Change in critical temperature (T_C_–T_C, 800 °C_) as a function of quenching temperature.

**Figure 3 materials-12-01156-f003:**
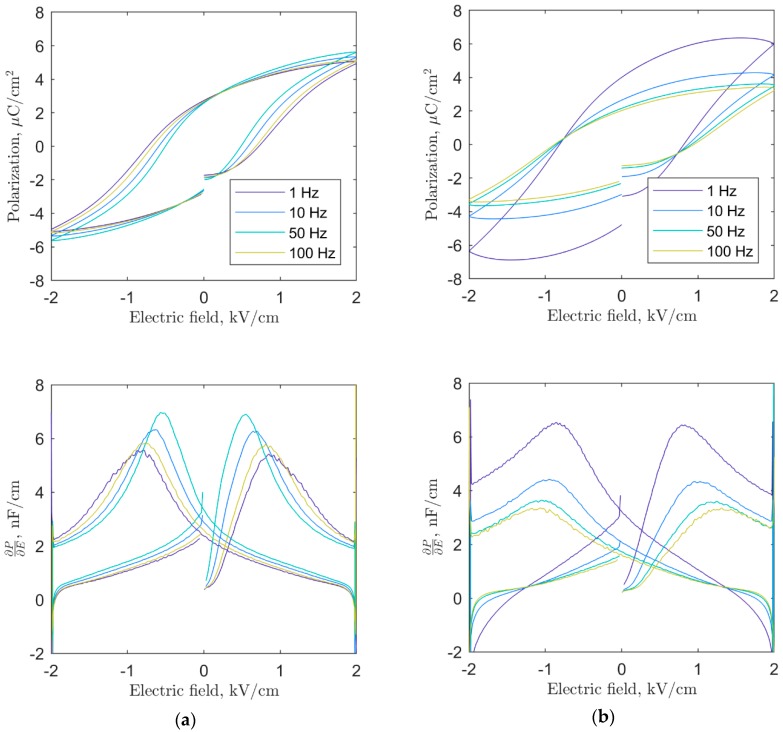
Ferroelectric hysteresis loop and current versus electric field for SBN 33 annealed at (**a**) 800 °C and (**b**) 1350 °C.

**Figure 4 materials-12-01156-f004:**
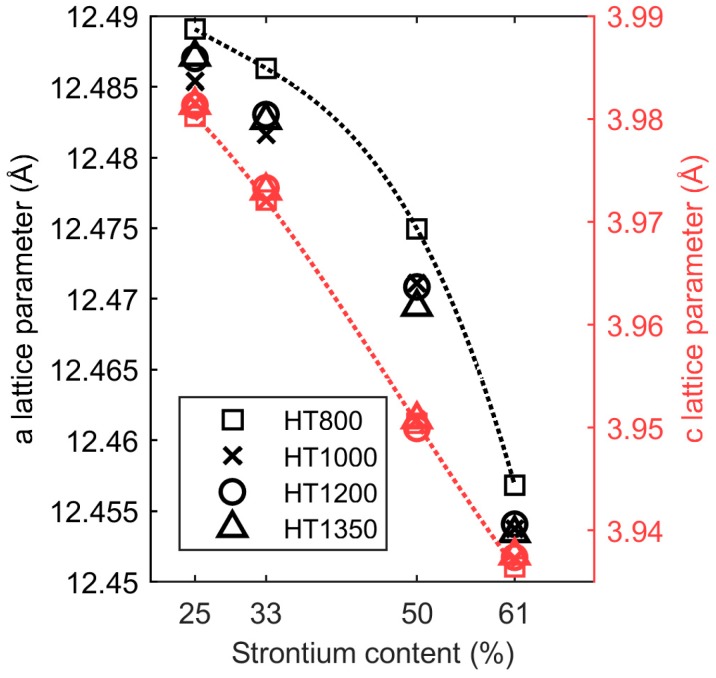
The *a* (black, left axis) and *c* (red, right axis) lattice parameters for all the compositions and heat treatments as calculated from the Rietveld refinements in TOPAS.

**Figure 5 materials-12-01156-f005:**
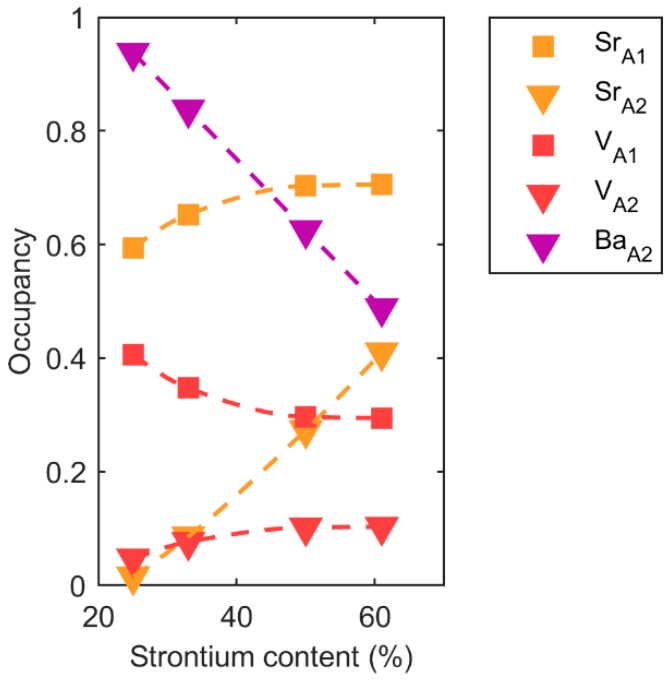
Occupancy of Sr, Ba, and vacancy on the A1 and A2 sites for the SBN ceramics heat treated at 800 °C.

**Figure 6 materials-12-01156-f006:**
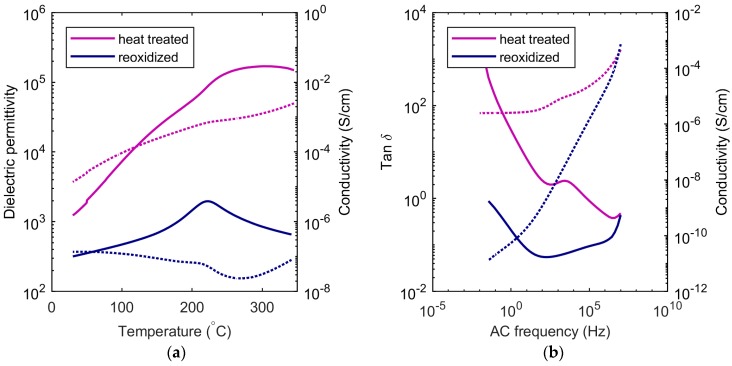
(**a**) Permittivity (solid lines) and AC conductivity (dotted lines) at 1000 Hz as a function of temperature for the SBN33 sample heat treated at 1350 °C before and after reoxidation. (**b**) tan δ (solid lines) and AC conductivity (dotted lines) for the same two samples as a function of frequency.

**Figure 7 materials-12-01156-f007:**
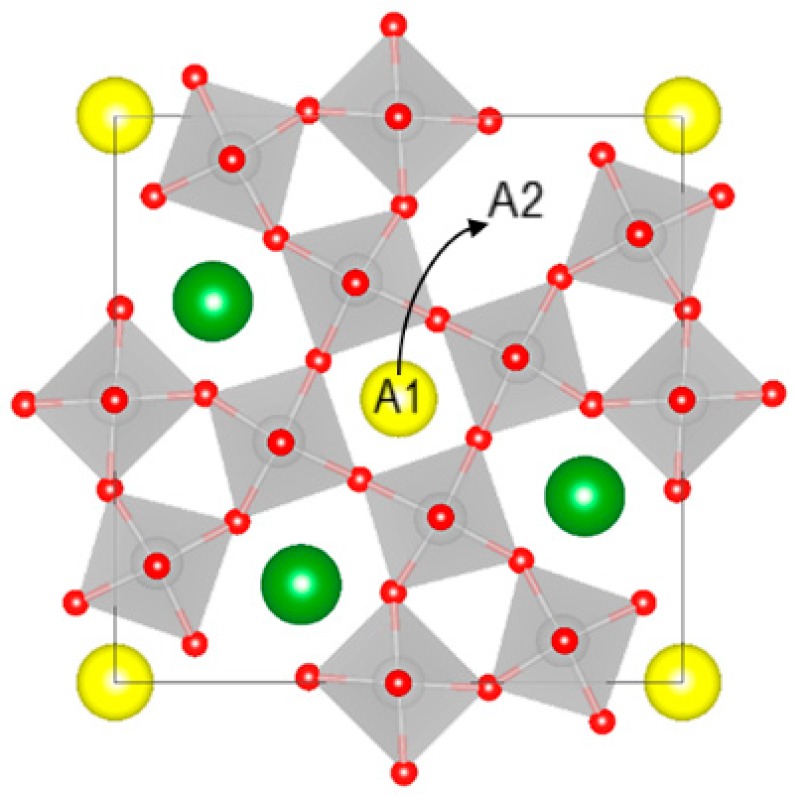
An example of cation/vacancy interchange; the strontium cation moves from A2 to A1. The figure was made using VESTA [17].

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
