# Peer review of "The Effect of Cation Disorder on Ferroelectric Properties of SrxBa1−xNb2O6 Tungsten Bronzes"

_materials, 2019, doi:10.3390/ma12071156_

Reviewer 1 Report

The authors have carried out a very nice study investigating the A-site disorder in SBN with different thermal histories. In general the study is thorough and the samples are analysed with a range of techniques to test the underlying hypothesis that high temperatures can drive site-swapping. The authors have been aware of other factors which could influence the results such as reduction due to high temperature processing. My only minor comment would be that it is well known that SBN exhibits varying degrees of relaxor behaviour as a function of Sr-content (and hence A2 occupancy); this observation was originally assigned to arise solely from A-site disorder analogous with perovskites (e.g. the text book example of the ferroelectric to relaxer behaviour in PbSc05.Ta0.5O3), however it is now more widely accepted that the relaxor behaviour in TTBs is influenced more by structural disorder manifested as incommensurations (in the ab plane). The authors should include some data and discussion of the frequency dependence of the permittivity as this is likely the real underlying cause of the non-linear change in a-lattice parameter and also the ferroelectric hardening. Regarding the latter, the observed ferroelectric hardening may also simply be due to the fact that the P-E data were collected further from Tc for the HT1350 sample than the HT800 (see fig 2a).

I recommend publication subject to these minor additions.

Reviewer 2 Report

Nowadays, TTB ceramics is a subject of high scientific interest as a lead-free ferroelectric, so the work presented can be considered quite relevant.
The experiments on the thermal quenching of tetragonal tungsten bronze (TTB) based on niobate ceramics and the investigations of the history of heat treatment influence on the structural and electrical properties of this material is quite interesting. However, some issues should be addressed to make the article more clear and the conclusions more justified.

Why ceramic samples of different compositions were sintered at different conditions (temperature and duration)? Since it is the effect of temperature that is being investigated, the explanation of this difference is important.

On the temperature dependences (Figures 1 and 2a), only the change in peak temperature is commented. The change in the absolute value of the dielectric constant is not explained at all, although it varies quite strongly in different directions for different samples.

The results of the reoxidation study are unclear. Why in figure 6a permittivity multiplied by 0.1? Why does conductivity increase with frequency? Maybe this is susceptance, not a conductivity?

Also there are some misprints and inaccuracies.

line 105 . "SIMENS" instead of "Siemens"

Figures 1, 2 and 6. Units of permittivity? Is it not dimensionless?
line 146, 148. Figures are referenced with letters "b" and "c", which are not placed in figure 3 itself.
line 234. "Valance" instead of "valence".

Author Response

Round  2

Reviewer 2 Report

No, I do not understand indeed.

At the bottom of the page 3 you have added the text about the frequency dependence of the permittivity. Where is this dependence? Does permittivity increase or decrease with the frequency? The only frequency dependence presented in the article is the figure 6b for the conductance and loss. Can you present the frequency dependence of the permittivity that you talk about? If you can, present it, please.

At the bottom of the page 6 and the top of the page 7 you write about tenfold increase in the permittivity. What are you talking about? One can look at the fig.2 and at the fig.6, and one can see that the permittivity dependencies of the SBN33 sample after the HT1350 treatment and after the reoxidization are nearly coinciding. You say that "the permittivity is for one of the sample is multiplied with 0.1 in order to be able to present the two data set in the same scale". Comparison of the fig.2 and fig.3 shows that the permittivity under question can be easily presented in the same scale.

And again I do not understand the fig.6b and the frequency dependence of the conductivity. What physical mechanism is responsible for the rise of the conductivity from 1e-11 to 1e-3 (by eight orders of magnitude)? Why it corresponds to the same rise of the frequency (from 1e-1 to 1e7, the same eight orders of magnitude)? May be you measure not the real conductivity, but the complex admittance. And below a frequency of 1e2 Hz the real conductance exceeds the imaginary susceptance for the sample before the reoxidization. For the reoxidized sample you have eliminated the real conductance and observed the susceptance only. Prove that I am wrong.

Author Response

Round  3

Reviewer 2 Report

OK, with conductivity it is now clear. The equivalent circuit of your sample is probably parallel connection of the capacitance (G1=i*omega*C), the equivalent conductance of dielectric loss (G2=tandelta*omega*C) and the leakage conductance (G3=G). So total conductance Gtot=G1+G2+G3, and Re(Gtot)=G+tandelta*omega*C. When you measure the sample with high leakage (HT1350) in frequency range (fig.6b), you observe leakage conductance at low frequency (up to 1e2 Hz) and dielectric loss conductance at high frequency. 1000 Hz is a frequency when the contributions of the both conductances are nearly equal.

After reoxidation you eliminate the leakage conductance due to the oxygen vacancies compensation, so you observe dielectric loss conductance only.

But now let's make clear the measurement of the permittivity. You claim that the occurence of the high leakage current leads to the two order rise of the permittivity. I do not understand this. Are you sure in your interpretation of the permittivity measurement in condition of the high leakage?

Additional doubts are caused by the rise of the permittivity with the frequency that you provided in the supplementary. Why it happens? In your own earlier work (Ref. 21) the permittivity decreased with frequency.
